# Is Allergic Rhinitis Related to Otitis Media with Effusion in Adults and Children? Applying Epidemiological Guidelines for Causation

**DOI:** 10.3390/cells14110805

**Published:** 2025-05-30

**Authors:** Ioannis Michael Vlastos, Natalia Tsotsiou, Mohannad Almomani, Maria Doulaptsi, Alexandros Karatzanis, Emmanuel Prokopakis

**Affiliations:** 1Department of Otorhinolaryngology, Evangelismos General Hospital, 10676 Athens, Greece; natalia.tsotsiou@yahoo.com (N.T.); moh199111@yahoo.com (M.A.); 2Department of Otorhinolaryngology, Heraklion University Hospital, 71500 Crete, Greece; mdoulaptsi@gmail.com (M.D.); akaratzanis@yahoo.com (A.K.); eprokopakis@gmail.com (E.P.); 3Laboratory of Translational Otorhinolaryngology Research, School of Medicine, University of Crete, 71500 Crete, Greece

**Keywords:** allergic rhinitis, otitis media with effusion, Bradford Hill criteria, causation

## Abstract

This review studies the potential relationship between the pathogenesis of allergic rhinitis (AR) and otitis media with effusion (OME) in both adults and children, applying the modified Bradford Hill criteria. While AR and OME are distinct conditions, several epidemiological and experimental studies suggest a significant association, primarily through allergic mechanisms such as Th-2 immune responses, Eustachian tube dysfunction, and inflammatory mediators in the middle ear. Given the substantial diversity and, in many instances, the “low quality” of related studies when assessed against the standards of modern evidence-based medicine, employing a structured framework like the modified Bradford Hill criteria is beneficial for investigating and establishing causality. This approach, which allows a wide range of diverse studies to be classified as direct, mechanistic, or parallel evidence, supports the notion that management of the allergic immune response may improve OME outcomes, although the inconsistencies among studies require further research. Despite current guidelines recommending against the use of antiallergic medications, the application of the above criteria suggests that proper diagnosis and treatment of allergic rhinitis should be strongly considered in adults and children with OME.

## 1. Introduction

Allergic rhinitis (AR) and otitis media with effusion (OME) are two prevalent conditions that share a complex and interrelated pathophysiology. AR is characterized by an IgE-mediated hypersensitivity reaction to inhaled or, less commonly, ingested antigens, which may provoke a range of allergic responses in the upper respiratory tract. This immune response is often triggered by common environmental allergens, leading to symptoms such as nasal congestion, itching, and sneezing. On the other hand, OME, also called seromucous otitis, is defined by the accumulation of fluid in the middle ear without acute infection. The condition is characterized by a range of clinical features, including persistent effusion in otoscopy; presence of B or C tympanogram; absence of ipsilateral acoustic reflex; and a conductive hearing loss greater than 25 dB at any one of the frequencies from 250 Hz to 4 kHz.

Numerous reviews on the role of allergy in otitis media with effusion (OME), in both adults and children, are available in the recent English literature, referencing various epidemiological and experimental studies. However, a structured approach to investigating and defining causality, such as the Bradford Hill criteria, has not yet been applied in these studies. Sir Austin Bradford Hill, in a widely cited pre-evidence-based medicine (EBM) framework for appraising evidence, proposed that several key factors should be considered before drawing conclusions about causation [1]. 

A recent reformulation of the criteria has been proposed within the context of evidence-based medicine (EBM), categorizing them into three groups: direct evidence, mechanistic evidence, and parallel evidence [2] (Table 1).

We believe that the modified Bradford Hill guidelines are valuable in addressing the potential relationship between allergic rhinitis (AR) and otitis media with effusion (OME), given the significant diversity and, in many cases, “low quality” of related studies when evaluated against “modern EBM” standards [3]. As the association between AR and OME is hypothesized, we have attempted to synthesize and evaluate the relevant studies using the modified Bradford Hill guidelines for causation.

A literature review has been conducted using the terms “allergy and otitis media with effusion” or “allergy and secretory otitis”, with additional related articles identified through references.

## 2. Direct Evidence

In epidemiological studies, it is crucial to ensure that potential confounders have adequately been accounted for. This means that the observed effect size must significantly exceed the combined effect of any plausible confounders (size of effect not attributable to plausible confounding). Regarding the relationship between allergy and otitis media with effusion (OME), numerous studies indicate a higher prevalence of allergies (not limited to allergic rhinitis) among patients with OME compared to the general population. This is particularly evident in cases such as eosinophilic otitis media and allergies manifesting as asthma. However, it is important to address the possibility of selection bias, as the physicians involved in these studies often had a specific interest in allergies (whether in the form of AR or other types like food allergies, atopy, or asthma). Nevertheless, a nationwide epidemiological study stated that children with allergic rhinitis were significantly associated with an increased risk of OME, even after adjusting for confounding factors such as age, gender, the levels of income for households, and household composition [4].

Another important consideration is the appropriate temporal and spatial proximity, or “temporality”, between cause and effect. Since causes precede their effects, an allergic reaction should trigger or worsen OME. This relationship has been explored in vitro as well as in vivo, both in human and animal studies.

In more detail, an in vitro experiment concluded that the interaction between dermatophagoides farinae (Der f) and lipopolysaccharide (LPS) in human middle-ear epithelial cells plays a significant role in the development of (OME) by inducing pro-inflammatory cytokines and mucin gene (MUC) [5].

Regarding human studies, nasal provocation tests were conducted in both atopic and non-atopic children and adults during the 1980s to investigate the relationship between allergic rhinitis (AR) and Eustachian tube (ET) obstruction.

In adults, studies have assessed nasal airway resistance using rhinomanometry and ET obstruction using tympanometry following nasal allergy provocation with histamine in both atopic and non-atopic individuals, finding a correlation between these factors. This test effectively distinguished individuals with AR from those without AR, a distinction that could not be made through nasal or skin responsiveness to histamine alone. The results suggest that atopic individuals exhibit greater hypersensitivity to histamine compared to non-atopic individuals, which may be a key pathogenetic mechanism leading to ET dysfunction and, potentially, the development of otitis media with effusion in the atopic group [6,7,8,9].

Eustachian tube function and middle-ear pressure (MEP) was also assessed in children with intranasal antigen provocation. Tympanometry evaluated changes in MEP, while Valsalva and Toynbee tympanometric tests were employed to study ET function. Additionally, changes in nasal airway resistance were measured using anterior rhinomanometry. The findings suggest that allergy plays a role in ET dysfunction and therefore in the development of OME [10,11].

Children with recurrent serous otitis media were also assessed for food allergies through skin prick tests, specific IgE tests, and food challenges. A food challenge was then conducted with the suspected allergens, while tympanometry was used to monitor middle-ear effusion throughout the pre-elimination, elimination, and challenge diet phases. A statistically significant correlation was found between food allergies and recurrent serous otitis media. The elimination diet improved symptoms in 86% of patients, while reintroducing the offending foods caused a recurrence in 94% [12].

Several animal experiments have been carried out to better understand and explain the pathophysiology of OME and its correlation with AR, as well as to explore potential treatment methods that could be effective in the management of OME in atopic patients. Juvenile rhesus monkeys were passively sensitized, and an allergic response was then induced by introducing an intranasal antigen or delivering it into the middle ear via the Eustachian tube. In some cases, Eustachian tube function was measured using tympanostomy tubes, with results supporting the connection between AR, ET dysfunction, and middle-ear pathologies. Histological evidence in other cases has suggested that a typical immediate Type 1 immune response occurs in the middle ear of sensitized animals, indicated by elevated IgE levels in both serum and effusion. Additionally, the presence of IgE-specific fluorescent mast cells in the middle ear suggested passive sensitization through circulating IgE. However, the detection of IgE-positive plasma indicated ongoing local synthesis of IgE as part of the continuing immune response [13,14].

Eustachian tube dysfunction has also been shown in rats that had been challenged transtympanically, as well as when intranasal allergy provocation tests were performed on mongrel dogs [15,16]. Another study investigated both the causal effect of allergy on OME in mice and the secondary impact of bacterial infection. The findings provided direct evidence of an allergic contribution to OME, as the Th-2 response, marked by strong expression of cytokines IL-5 and IL-13, was prominent. Additionally, the Th1 response (IL-2 and IFN-γ) to lipopolysaccharides was more pronounced in the allergic group. This suggests that allergy-related otitis media can be significantly exacerbated by an inflammatory reaction due to a bacterial infection [17].

However, not all animal experimental studies published in the current literature support the relationship between allergy and OME. There are studies suggesting that nasal allergies have little direct impact on the middle ear and ET [18,19]. More specifically, histological examination of the ET and tympanic cavity in sensitized guinea pigs revealed allergic changes when the antigen was introduced directly into the tympanic cavity. However, no such changes were observed when the antigen was administered intranasally.

Additionally, tympanometry and otomicroscopy were conducted on passively sensitized juvenile rhesus monkeys following intranasal and ET challenges. On the final day of the challenge, tympanocentesis was performed, and the middle ear was examined histologically. The results did not show the initiation of an inflammatory reaction in the middle ear or the induction of OME [20]. These results are in line with experiments in guinea pigs. IgE-mediated allergic reactions in the mucous membranes of the nose, nasopharynx, and ET, along with a high density of mast cells observed at the pharyngeal orifice of the ET, were indicative of a chronic disease state rather than a direct cause of otitis media with effusion (OME) [21,22]

A study exploring the pathophysiology of eosinophilic otitis media has also been published in the literature. In rats with eosinophilic otitis media, the pathogenesis of the condition was further explored by dissecting the temporal bones. Examination revealed that the epithelium of the ET was swollen and thickened, with cilia arranged in a disorderly manner and partially detached. There was significant infiltration of eosinophils into the submucosal layer of the ET, accompanied by mast cell degranulation. Scanning electron microscopy showed that the cilia were aggregated along the entire length of the ET. These findings suggest that the disruption of normal ET function plays a critical role in the occurrence and progression of eosinophilic otitis media [23].

In addition to investigating the causal relationship between AR and OME, studies have explored potential treatments for OME in animal models to determine whether addressing the allergic immune response in the middle ear could prevent or cure OME associated with allergies. Proposed treatments include oral azelastine hydrochloride and transtympanic administration of antihistaminic drugs, as well as immune modulatory oligonucleotides, soluble interleukin-4 receptors (sIL-4R), and interleukin-5 antibodies (IL-5Ab). These treatments have shown favorable outcomes in managing OME in allergic animals, supporting the relationship between allergy and OME [24,25,26,27].

Subsequent to the above, the final “direct criterion” refers to “dose responsiveness” (or “biologic gradient”). This criterion is observed when changes in the outcome correlate with variations in the intensity of the intervention or the cause.

If we hypothesize that allergic rhinitis leads to or worsens OME, then treating the underlying allergic condition should also help resolve the OME. In this context, several studies have been published involving both children and adults with OME who were treated with antihistamines, corticosteroids, immunotherapy, or food elimination diets. The majority of these studies suggest that addressing the allergic immune response can yield positive outcomes for managing OME [28,29]. For instance, treating AR led to a significant improvement in hearing thresholds in children suffering from hearing loss of up to 33 dB [30]. Additionally, children treated with inhalant medications and who followed food elimination diets for one year showed significant decreases in OME recurrence and need for myringotomy [31]. 

Similar results were observed where patients were treated with both immunotherapy and food elimination diets, resulting in a complete resolution of symptoms and no recurrence over a three-year follow-up [32]. Newer antihistamine medications appear to have positive effects on OME without the adverse effects associated with older H1 antagonists [33]. In addition, hydroxyzine pamoate showed favorable outcomes when administered through grommets after tympanostomy [34]. Contrary to the above, there are studies showing an ineffectiveness of antihistamines and decongestants after 1–3 months of follow-up of OME [35,36,37,38]. As is explained further in the discussion, current guidelines on avoidance of these medications are based mostly on these older trials. In a more recent study, an antihistamine inhalant medication failed to demonstrate a statistically significant efficacy for AR in curing OME. However, even in this study, the medication appeared to contribute to OME treatment by improving nasal symptoms and was finally recommended for treating AR in patients with OME [39].

The importance of allergy treatment for the resolution of OME is also shown from studies on surgical interventions. In one of our previous studies, it was evident that a history of allergy was a contraindication for laser assisted tympanostomy without tympanostomy tube insertion [40]. This was also the case with laser Eustachian tuboplasty, which was not effective in the presence of allergic rhinitis (AR) or laryngopharyngeal reflux [41]. These findings suggest that OME is, at least in part, an immune-mediated allergic disease and that patients with OME should be considered for thorough allergy evaluation and treatment, as many respond well to immunotherapy. This approach is particularly beneficial for children with seasonal recurrences of middle-ear effusion, a history of allergic diseases in infancy or early childhood, and a family history of allergy [42]. 

Regarding the treatment of eosinophilic otitis media, targeting the allergic immune response with biological agents against IL-5, IL-4, or IL-13 proved effective in several studies. Additionally, one study reported favorable outcomes with triamcinolone acetonide infusion into the mesotympanum. However, in all cases, the therapy had minimal effect on patients with granulation or thickening of the middle-ear mucosa [43,44,45].

## 3. Mechanistic Evidence

Increasing evidence suggests that allergic mechanisms, particularly those involving a Th-2-type immune response, may play a significant role in a subset of OME cases. The analysis of cytokine levels in adults with OME, including IL-2, IL-4, IL-5, IL-10, IL-12, and interferon (IFN)-gamma, reveals distinct patterns associated with otitis media with effusion (OME) and allergic conditions. Notably, IL-4 levels were significantly higher in patients with allergic rhinitis compared to those without. Additionally, IL-10 was found at increased levels in mucoid-type middle-ear effusions compared to serous types, suggesting its role in influencing effusion viscosity. Regardless of allergic status, IL-12 may play a critical role in OME pathogenesis by impacting the production of IL-2 and IFN-gamma, while IL-4 may contribute to the immunologic profile of adults with AR [46].

The involvement of allergic mechanisms in OME is further supported by the presence of antigen-specific IgE and elevated levels of eosinophils and eosinophil cationic protein (ECP) in the middle-ear effusion of allergic individuals. It is shown that IgE, particularly against inhalant and bacterial antigens, can be locally produced in the middle-ear mucosa, potentially contributing to the severity of OME in allergic patients. This localized allergic response is also reflected in the higher expression levels of cytokines associated with Th-2-driven inflammation, such as IL-4 and IL-5, in the middle-ear fluid of allergic children [47,48,49,50,51,52,53,54,55].

Additionally, the concept of the “united airway” has been proposed, suggesting that the middle-ear mucosa may participate in the same allergic inflammatory processes affecting the upper respiratory tract. Evidence from studies showing similar allergic inflammation profiles in both the middle ear and nasopharynx in atopic patients supports this hypothesis. This idea is reinforced by findings of elevated levels of Th-2 cytokines, such as IL-4, IL-5, and IL-13, and the presence of mast cells and their mediators, including tryptase, in the middle-ear effusion, which indicates that allergic inflammation could contribute to the persistence of OME [56,57,58,59,60,61].

The role of microRNAs (miRNAs) in the molecular mechanisms of ear effusion formation has recently been highlighted. miRNAs, known to regulate allergic responses, show differential expression in allergic versus non-allergic patients with OME. Specifically, miR-320e has been found to be significantly decreased in allergic children with OME, suggesting that it may play a role in the pathophysiological mechanism of effusion formation in these patients. However, the exact role of miR-320e and other miRNAs in OME pathology requires further investigation [62].

The role of mechanical nasal mucosal swelling in the development of OME has been debated. It is well established that AR can lead to ET dysfunction through the swelling of the nasal mucosa, which can obstruct the ET and impair its ability to equalize pressure in the middle ear. This obstruction is thought to contribute to the development of OME by promoting the retention of fluid within the middle ear. However, mechanical swelling alone may not fully explain the association between AR and OME. Instead, the allergic inflammation itself, rather than just the physical obstruction caused by swelling, might be a more significant factor. Inflammatory mediators such as cytokines (e.g., IL-4, IL-5) and chemokines (e.g., RANTES) that are elevated in allergic individuals could directly contribute to the pathogenesis of OME by promoting mucosal inflammation and fluid accumulation in the middle ear [63,64,65,66,67].

Atopic individuals, characterized by a heightened Th-2 immune response, may also exhibit increased sensitivity to bacterial infections in the middle ear. Research suggests that atopic patients respond differently to bacterial and viral products of acute inflammation, owing to the presence of primed inflammatory cells. For instance, the presence of neutrophils, a key component of the immune response, is markedly elevated in the effusions of atopic patients with chronic OME. This suggests that atopic individuals might experience more intense or prolonged inflammatory responses when exposed to bacterial pathogens, leading to a more severe or persistent course of OME [68].

However, not all studies fully support the role of allergy in OME. Some research has failed to demonstrate a strong association between IgE-mediated allergic reactions and the presence of OME, suggesting that allergy may not be a significant factor in all cases. For instance, studies using techniques like radioimmunoassay to measure IgE levels in middle-ear effusion have not consistently shown elevated IgE concentrations in patients with OME, leading some researchers to question the role of atopy as a major causative factor. For this reason, some authors argue that investigating allergy as part of the diagnostic process for children with recurrent otitis media with effusion (OME), who lack a history of allergies based on clinical evaluation, family history, or laboratory tests, is likely to be nonproductive [69,70,71,72,73]. In conclusion, further research is needed to clarify the exact contribution of allergic inflammation to OME and to identify which patients may benefit from targeted allergy management as part of their treatment plan.

## 4. Parallel Evidence

Replicability (or consistency, as initially defined by Bradford Hill) refers to obtaining similar results when the same intervention is applied to comparable populations, using the same outcome measure. Numerous studies point to a significant association between allergic rhinitis and an increased risk of OME in children or adult-onset OME (AO-OME) [74,75,76,77,78,79,80,81,82,83,84,85,86,87,88,89,90,91].

For instance, in a study by Byeon et al., children with allergic rhinitis were found to have double the risk of developing OME compared to non-allergic children (OR = 2.04) even after adjusting for other confounding factors [4]. Similarly, research by Norhafizah et al. identified that 80.3% of children with persistent OME had a coexisting diagnosis of allergic rhinitis [30]. These findings suggest that AR may contribute to OME through immune responses in the middle ear, such as local IgE production or eosinophilic inflammation [92,93,94].

The presence of allergic markers such as elevated IgE levels or eosinophil cationic protein in the serum and effusion of patients with OME, as highlighted in studies by Bernstein and Hurst, strengthens the argument for a specific immunological link [95]. Furthermore, a large dataset representing 1,491,045,375 pediatric visits validated the association between allergic rhinitis (AR) and otitis media with effusion (OME) and highlighted that age plays a crucial role in modifying this relationship. Specifically, in children 6 years of age or older, the presence of AR significantly increased the odds of developing OME, Eustachian tube dysfunction (ETD), or tympanic membrane retraction (TMR) [96]. As a result, these studies suggest that managing allergic rhinitis could play a key role in preventing or treating patients with recurrent OME.

Currently, with the development of genetic analyses, Mendelian randomization studies are being utilized to minimize the confounding and bias often seen in observational studies. These studies use genetic variations as natural experiments to assess the causal relationship between a risk factor and an outcome. Although these studies ensure temporality, because genetic variants are present from birth, preceding the development of an outcome, such as otitis media, they offer mostly indirect or parallel evidence. There is one Mendelian randomization study for allergic rhinitis and otitis media, which revealed a significant causal effect of AR on nonsuppurative otitis media [97].

On the other hand, other observational studies present conflicting evidence, challenging the idea that allergies are directly responsible for otitis media with effusion. For example, some research has failed to find significant differences in the prevalence of OME among allergic and non-allergic children, suggesting that other factors, such as viral infections, may play a more dominant role in the development of OME [63,98,99,100,101,102,103,104,105,106,107,108]. In another study, atopic children were observed during seasonal exposure to ragweed pollen. The findings suggest that children with ragweed hay fever experience ET dysfunction during natural pollen exposure. However, while seasonal allergy can induce ET dysfunction, it alone did not seem sufficient to cause middle-ear effusion [109].

Overall, the relationship between AR and OME remains a subject of debate within the scientific community. While there is substantial evidence from studies supporting a connection between the two conditions, particularly in patients with clear allergic triggers, other research highlights inconsistencies and suggests that more work is needed to understand the full scope of this relationship. 

## 5. Discussion

The pathogenesis of OME is complex, involving factors such as ET dysfunction, infections, and inflammation. Among them, the role of allergic mechanisms, particularly those involving T(H)2-type immune responses is being increasingly highlighted as a contributing factor. Allergic rhinitis, with its associated inflammatory mediators and immune responses, may influence the development of OME by affecting the middle ear’s mucosal environment. This relationship is further complicated by varying prevalence rates and the presence of additional risk factors, such as socioeconomic status, smoking, and allergy history in the family.

Despite the concept of “one airway, one disease”, which integrates the middle ear within the broader context of respiratory allergies [110], the precise role of allergy in the development and persistence of OME remains unclear. Research indicates that while allergens might contribute to the inflammatory substrate in OME similarly to other allergic conditions like asthma, the degree of their involvement varies. This variability underscores the need for further investigation into the interplay between allergic rhinitis and OME for improved clinical management strategies.

In a previous study using the same structured framework to explore a potential causal relationship between allergic rhinitis and chronic rhinosinusitis, we analyzed pediatric and adult studies separately, given the notable differences in the number, quality, and methodology of available studies for each group. In the current study, however, we did not perform age-based separation. Although OME is significantly more prevalent in children—and there are important differences in pathophysiology and clinical management across age groups—we identified several studies in both children and adults that could be classified as providing direct, mechanistic, or parallel evidence for a relationship between OME and allergic rhinitis. It is worth noting that current guidelines for the management of OME are largely based on pediatric studies, reflecting the higher prevalence of the condition in this population.

This review employs the modified Bradford Hill criteria to assess the causality of the speculated association. Multiple observational and experimental studies indicate that AR, through mechanisms such as ET dysfunction and local inflammatory responses, can contribute to the development of OME. The presence of allergic markers, such as elevated immunoglobulin E (IgE) levels and Th-2-driven cytokines (IL-4, IL-5), in the middle-ear effusions of patients with AR, strengthens the hypothesis that allergy-induced inflammation plays a role in OME pathogenesis.

However, not all studies support this relationship consistently. Some experimental models, particularly animal studies, fail to show direct causation between nasal allergy and middle-ear pathology, while others indicate that mechanical factors such as mucosal swelling may exacerbate, but not solely cause, OME. Updated guidelines provide a strong recommendation against the use of antihistamines and steroids for treating OME [111]. Although steroids are effective, their duration of action is short-term and it does not justify the risk of its administration. Regarding antihistamines with or without decongestants, the recommendation is based on a Cochrane review. In this review, the pooled data demonstrated no benefit and some harm from the use of antihistamines and decongestants [38]. However, this conclusion was mostly based on three studies [35,36,37] from the early 1980s. Thus, they had utilized older-generation antihistamines, which have a significant number of adverse events. In addition, and probably of the greatest importance, is that they have some methodological issues, such as the small number of patients that were followed for over one month.

Several reviews and meta-analyses have explored allergy as a risk factor for OME development but have not applied the Bradford Hill criteria—a framework that still provides a rigorous means of classifying a wide range of studies. Previous recommendations against using antiallergic medications are now being revalidated: a recent EAACI Position Paper [112] advocates allergy testing and topical steroids in children, while adults may benefit from an even broader array of pharmacological treatments. Moreover, an increasing number of reports now support an association between allergic reactions and Eustachian tube dysfunction [113].

All of this underscores a sustained and vigorous interest in the field. Beyond this, the complex interplay between allergic and infectious inflammation has begun to emerge as a key research area. Epidemiological data collected during the COVID-19 pandemic [114] suggest interactions between allergy-driven cytokines and humoral immunity. Pathophysiologically, children with allergic rhinitis show high human rhinovirus (HRV) detection rates, which correlate positively with symptom severity [115] , and early-life infections appear to raise later AR risk [116]. Pharmacologically, there is growing support for immunomodulators such as pidotimod [117,118] and for certain herbal remedies [119].

## 6. Limitations

The Bradford Hill criteria were originally developed to assess environmental and occupational risks. Their modification allows them to be applied to complex, multifactorial conditions such as chronic diseases. However, there are no clear thresholds or metrics defining how each criterion should be satisfied. In the context of exploring a causal relationship between allergic rhinitis (AR) and otitis media with effusion (OME), restricting inclusion to studies with explicit and consistent definitions—particularly regarding the duration of effusion—would have excluded the majority of available research. This is especially true for experimental studies that provide evidence of mechanistic action or temporality. In addition, direct evidence relies heavily on animal studies using models that may have little to do with the induction of allergic rhinitis in children.

## 7. Future Prospects 

Given the limited published evidence, current management strategies for OME in the context of allergic rhinitis rely heavily on expert consensus rather than high-quality data. It is particularly important to elucidate the mechanisms by which IgE-mediated allergic responses contribute to middle-ear pathophysiology, including Eustachian tube dysfunction, mucosal inflammation, and local immune modulation. In addition to classical antiallergic treatments—such as topical corticosteroids and antihistamines—emerging approaches like allergen immunotherapy and immunomodulatory agents represent a promising area of research in both pediatric and adult populations. These therapies may not only target systemic allergic pathways but could also have localized effects on middle-ear inflammation and immune balance. Future research should prioritize well-designed randomized controlled trials, as well as real-world evidence studies, to address existing knowledge gaps. Multidisciplinary approaches integrating allergology, otolaryngology, and immunology will be critical for developing more precise diagnostic tools and personalized treatment strategies. Ultimately, such efforts may lead to more effective and targeted management of OME in patients with underlying allergic rhinitis, improving both short and long-term clinical outcomes.

## 8. Conclusions

While the association between AR and OME is supported by numerous studies, the variability in findings highlights the need for additional research to confirm causality and improve treatment protocols for OME related to allergic response. Nevertheless, our approach with the application of the modified Bradford Hill criteria supports the notion that management of the allergic immune response may improve OME outcomes despite current guidelines recommending against the use of antiallergic medications. 

## Figures and Tables

**Table 1 cells-14-00805-t001:** Applying the modified Bradford Hill guidelines on the evidence in favor of and against allergy being a predisposing factor for OME.

Direct	Size of effect not attributable to plausible confounding	Increased risk of OME in children with AR after adjusting for confounding factors Risk of selection bias in studies regarding food allergies, atopy, asthma, or eosinophilic otitis media
	Appropriate temporal and/or spatial proximity (cause precedes effect, and effect occurs after a plausible interval; cause occurs at the same time as the intervention) (strength)	Experimental studies with allergy provocation in children and adults indicate potential relationship between OME and AR, possibly due to ET dysfunction Some animal experimental studies fail to prove the connection
	Dose responsiveness and reversibility (temporality)	An array of studies suggests that treating the allergic immune response can lead to positive outcomes in the management of OME
Mechanistic	Evidence for a mechanism of action (biologic, chemical, mechanical) (biologic gradient, biologic plausibility)	Elevated levels of Th-2 cytokines (e.g., IL-4, IL-5, IL-13) and the presence of mast cells and mediators like tryptase have been detected in middle ear effusions, indicating an allergic inflammatory response in OME The “united airway” concept suggests that allergic inflammation affecting the upper respiratory tract may also impact the middle ear, contributing to the development of otitis media with effusion The role of mechanical swelling in Eustachian tube dysfunction remains controversial
Parallel	Coherence (coherence) Replicability (consistency) Similarity (analogy)	Inconsistent diagnostic criteria; only recently have more explicit criteria been used in epidemiological studies Numerous studies highlight a significant association between allergic rhinitis and an increased risk of OME in children and adults, although this is not consistent in all of the studies

OME; otitis media with effusion, AR; allergic rhinitis.

## Data Availability

No new data were created or analyzed in this study.

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
