# Peer review of "Is Allergic Rhinitis Related to Otitis Media with Effusion in Adults and Children? Applying Epidemiological Guidelines for Causation"

_cells, 2025, doi:10.3390/cells14110805_

Round 1
Reviewer 1 Report
Comments and Suggestions for Authors
The authors present a timely and thoughtful synthesis of evidence examining the potential causal link between allergic rhinitis (AR) and otitis media with effusion (OME).
Here are some review comments:
1) Page 1, Line 14
The abstract refers to the “modified Bradford Hill criteria” but does not briefly explain what these are or why they are particularly suited to this topic.
2) ABSTRACT: Page 1, line 27
It would be helpful to clarify whether the strength of the association between AR and OME differs between pediatric and adult populations, given the notable differences in pathophysiology and clinical management across age groups.
3) INTRODUCTION: Page 1, lines 31-40
The definitions of AR and OME lack sufficient clinical detail. Given their central importance to the review, more precise and comprehensive descriptions are needed to support reader understanding.
4) ALL TEXT:
Citations are presented inconsistently (e.g., “6-7-8-9” instead of “6–9”).
5) DISCUSSION:
Consider discussing the potential role of Pidotimod as an immunomodulatory treatment in OME, especially in allergic or infection-prone patients.
6) DISCUSSION:
Consider adding a note on how reduced community exposure, such as during COVID-19 lockdown, might temporarily lower the incidence of OME and allergic exacerbations by limiting contact with common allergens and pathogens (I suggest citing the following article: doi:10.1177/0194599820987458)
7) DISCUSSION: Page 8, line 295
I suggest adding a detailed section on study limitations.
8) DISCUSSION: Page 8, line 295
I suggest inserting a “Future Prospects” section after the limitations.
9) DISCUSSION: Page 8, line 296
I suggest creating a separate "Conclusion" section.
Author Response
we would like to thank the reviewer for the nice words and the constructive comments.
please attached find our reply

Reviewer 2 Report
Comments and Suggestions for Authors
hello thank you for submitting your paper concerning relationship between allergic rhinitis and otits media with effusion
The paper is interesting May be to avoid any misunderstanding it would be nice to change the titile and to replace otitis media with effusion with seromucous otitis media This avoids any confusion with an acute otitis media which is associated to an infection
By the way the topic is interesting and your papzr well organized
practically we can say that patients with allergic rhinitis have a propension to have otitis media with effusion . This is more prevalent in children than in adults
But we cannot say that all patients particularly adults with otits media with effuison must be tested for allergy
In the future further studies are necessary to identify which parameters is more prevalent to observe such association
Author Response
We would like to thank the reviewer for the positive feedback and kind words.
please attached find our response

Reviewer 3 Report
Comments and Suggestions for Authors
There is a very recent position paper published on allergic rhinitis and OME (https://doi.org/10.1111/all.16554) and quite good review in the World Allergy Organization Journal (10.1016/j.waojou.2023.100860). Apart from this there have been comprehensive recent overviews. None have used the Bradford Hill criteria which is still a good framework and strength of this paper.
The direct evidence section relies heavily on animal studies using models that have little to do with the induction of allergic rhinitis in children. Some recognition of this would be recommended.
Line 144. It would be better to state use "animals" instead of "subjects " in the sentence ending in "outcomes in managing OME in allergic subjects, supporting the relationship between allergy and OME". This makes it clear that the subjects were animals.
There is little about possible interactions or co-occurrence of infections and OME. Papers that could be helpful for the rhinitis side are DOI: 10.1186/s12887-023-03870-0 and DOI: 10.1128/spectrum.03853-23
Editing of the references is needed
Author Response
we would like to thank the reviewer for his constructive comments
please attached find our response

Round 2
Reviewer 1 Report
Comments and Suggestions for Authors
The authors have clarified and improved the manuscript.